Interpopulation morphological differences and sexual dimorphism of Dekay’s brownsnake (Storeria dekayi) along a rural–urban gradient

Huang Tianqi tianqi.huang@rutgers.edu 1 2
Morin Peter J. 1
Ruane Sara 2
1 Department of Ecology, Evolution, and Natural Resources, Rutgers, The State University of New Jersey , New Brunswick , NJ , United States of America
2 Life Sciences Section, Negaunee Integrative Research Center, Field Museum of Natural History , Chicago , IL , United States of America
Manjarrez Javier
Electronic publication date: 2025 Jun 11
Publication date: 2025
Volume: 13
Electronic Location ID: e19439
Received 2025 Jan 14; Accepted 2025 Apr 16
Copyright: ©2025 Huang et al.
Copyright year: 2025
Copyright holder: Huang et al.
License: This is an open access article distributed under the terms of the Creative Commons Attribution License, which permits unrestricted use, distribution, reproduction and adaptation in any medium and for any purpose provided that it is properly attributed. For attribution, the original author(s), title, publication source (PeerJ) and either DOI or URL of the article must be cited.
License URL: https://creativecommons.org/licenses/by/4.0/

Keywords: Geometric morphometrics, Head morphology, Phenotypic shift, Urbanization

Funding: Rutgers University-Newark Field Museum of Natural History This work was supported by funds granted to Sara Ruane via Rutgers University-Newark and the Field Museum of Natural History (Grainger Bioinformatics Center, the Negaunee Integrative Research Center, and the Women’s Board). There was no additional external funding received for this study. The funders had no role in study design, data collection and analysis, decision to publish, or preparation of the manuscript.

==============================
In response to the surge of urbanization in the modern era, many organisms have undergone various changes, such as the shift of their morphological traits to face the challenges brought by this drastic environmental transformation. Rapid adaptive evolution in the morphology of urban-dwelling organisms has been documented in a broad array of taxa, such as lizards and birds, by comparing urban populations with their nonurban counterparts. However, relevant studies concerning more elusive and secretive organisms that also occur in both natural and urbanized habitats (e.g., snakes), are still lacking. Snakes lack appendages, which are often the trait of interest in other morphological studies, but factors such as head shape play a critical role in snakes, as it determines the prey size of these gape-limited predators. In this study, we apply both linear and geometric morphometric analyses to examine interpopulation morphological differences and sexual dimorphism in a small, semi-fossorial snake, Dekay’s brownsnake (Storeria dekayi). We focus on head shape in six different populations across the rural-urban gradient in New Jersey and New York, USA. We find evidence of increased morphological divergence and decreased sexual dimorphism in populations inhabiting more urbanized areas. Our study suggests the occurrence of an adaptive morphological shift in this common species in the urban environments, and lays the path for further investigation of urban adaptation in snakes and similar secretive species.

Introduction

Environmental changes can influence the morphology of organisms (Kappelman, 1988; Grant & Grant, 1989; Losos, Warheitt & Schoener, 1997). With rapid urbanization occurring in recent decades, drastic morphological shift would be expected on organisms living in the novel urban environments. Through comparing urban and nonurban populations, local adaptation in urban habitats has already been documented in Anolis lizards, which have become model organisms for studies on adaptive evolution in response to novel urban selective pressures. For instance, urban brown anoles (Anolis sagrei) have evolved larger hindspans in response to novel artificial perching surfaces (Marnocha, Pollinger & Smith, 2011). Similarly, urban populations of crested anole (Anolis cristatellus) display longer limbs and more lamellae than their rural conspecifics (Winchell et al., 2016). By examining the claw morphology of five Anolis species, Falvey et al. (2020) discovered that urban Anolis tend to have shorter and more robust claws. All the aforementioned changes in Anolis limb morphology allow those small reptiles to navigate the concrete jungles effectively, making them successful urban dwellers. These findings suggest that morphological adaptation to urban habitats is likely to occur in other groups of animals, many of which have not been investigated in this context.

As opposed to genetic variation, which is often reduced by urbanization due to habitat fragmentation (Rubin et al., 2001; Delaney, Riley & Fisher, 2010; Munshi-South, Zak & Pehek, 2013; DeCandia et al., 2019), morphological variation is frequently found to be increased by urbanization in a wide spectrum of taxa (e.g., bees: Eggenberger et al., 2019; mosquitos: Multini, Wilke & Marrelli, 2019; lizards: Falvey et al., 2020; birds: Thompson et al., 2021). However, many details are yet to be investigated in this relatively new field of morphology research among urban taxa. For example, in most relevant studies, comparisons are only drawn between pairs of urban and nonurban populations without considering the presence of a rural–urban gradient with a broad range of urbanization levels. Furthermore, by comparing different urban and nonurban populations, studies often fail to take factors contributing to intrapopulation variation, like sexual dimorphism, into account.

Urban environments have the potential to alter inhabitants’ magnitude of sexual dimorphism. On islands, it is known that locally specific environmental features can alter the process of resource extraction and allocation for the sex with more variable phenotypes, leading to differentiated yet pronounced expressions of sexual dimorphism among different insular populations (Cox & Calsbeek, 2010; Sacchi et al., 2015; Muraro et al., 2022). Given their unique human-altered features, urban environments are thought to increase or favor mechanisms that increase sexual differences in a similar fashion (Chulisov, Konstantinov & Vongsa, 2019). Despite that, by comparing plumage colors of different bird species, studies have given contradicting results regarding the relationship between urbanization and the magnitude of sexual dimorphism (Kark et al., 2007; Croci, Butet & Clergeau, 2008; Leveau, 2019). Additionally, with such studies predominately comparing the magnitude of sexual dimorphism between different species, more work is necessary to test the relationship between urbanization and sexual dimorphism intraspecifically.

Though not as evident as that seen in birds, sexual dimorphism is also widely exhibited in snakes (Shine, 1991a). Sexual dimorphism in snakes is shaped by various aspects of their natural history, such as their sexually differentiated reproductive biology (King, 1989; Shine, 1994; Bonnet et al., 1998; John-Alder, Cox & Taylor, 2007) and foraging strategy (Houston & Shine, 1993; Vincent et al., 2006a). Although snake populations are experiencing a worldwide decline under anthropogenic pressure (Reading et al., 2010), some species still manage to thrive in urbanized environments, and exhibit differentiated natural history from their nonurban counterparts (Zappalorti & Mitchell, 2008; Pattishall & Cundall, 2009; Sullivan, Leavitt & Sullivan, 2017; Wolfe, Bateman & Fleming, 2018). Therefore, urbanization would also be expected to cause changes in the morphology and magnitude of sexual dimorphism of urban-dwelling snakes. In one study, urban European grass snake (Natrix natrix) shows less sexual size difference compared to suburban and nonurban populations owing to the lack of food source in the city (Bury & Zając, 2020). However, besides the work by Bury & Zając (2020), to the best of our knowledge, no literature to date has further investigated the relationship between morphology, sexual dimorphism, and urbanization in snakes.

Dekay’s brownsnake (Storeria dekayi) is a small, semi-fossorial natricid snake found throughout most parts of eastern and central North America (Ernst & Ernst, 2003). It inhabits a wide variety of terrestrial habitats, but is most commonly found in anthropogenically disturbed and urbanized environments, such as abandoned quarries and urban greenspaces (Noble & Clausen, 1936; Ernst & Ernst, 2003; Zappalorti & Mitchell, 2008; Huang, Morin & Ruane, 2025). Storeria dekayi is a dietary specialist that feeds almost entirely on terrestrial annelids and gastropods (Gray, 2013; Virgin & King, 2019). Across its range, geographical differences of S. dekayi regarding its morphology are mainly demonstrated by slight pigment variations (Ernst & Ernst, 2003). However, S. dekayi displays sexual dimorphism, as males generally have a smaller body size but a larger head size than females, a rare condition as females commonly exceed males in both body and head size among snakes (Shine, 1991a; King, 1997). Here, we present a study that examines the morphology of S. dekayi across sites with different levels of urbanization in New Jersey and New York, USA. Our goal is to investigate the extent of morphometric differentiation and to evaluate sexual dimorphism in S. dekayi across the rural–urban gradient with respect to its head shape, an important trait of this gape-limited predator (Shine, 1991b; King, 2002; Vincent et al., 2006b). Because little is known about the morphological change of many secretive yet common taxa in the cities, we hope this study can help enrich the existing natural history literature on such taxa, and suggest viable hypotheses on the effect of urbanization on wildlife morphology.

Materials & Methods

Between 28 September 2017 and 23 June 2021, we sampled mostly live S. dekayi from five different locations across New Jersey: Rutgers University-Newark (NW), Rutgers University-New Brunswick (NB), Perth Amboy (PA), Fort Lee (FL), and Maurice River (MR) (Fig. 1). Snakes were primarily sampled by uncovering both artificial and natural shelters in potential habitats. Some snakes were also acquired through opportunistic visual encounters. Voucher specimens collected from Van Cortlandt Park (VC), New York in the herpetological collection of the American Museum of Natural History (AMNH) were also included as an additional population in this study. To quantity the level of urbanization, we calculated the percent imperviousness of each site using the NLCD 2019 Percent Developed Imperviousness (CONUS) dataset (Dewitz & US Geological Survey, 2021). Each site was defined as a circular area occupied by a distinct S. dekayi population, with the coordinate centroid of all occurrence records of that population as the center and a radius of two km. Because S. dekayi has a small home range (Ernst & Ernst, 2003), a radius of two km should be sufficient to encompass the home range of all members in a population. Because coordinates of the VC specimens were not provided, the centroid of Van Cortlandt Park was used for the analysis.

Figure 1 Locations of the six Storeria dekayi populations used in this study.

Most snakes sampled in the field were measured, photographed, and then released on-site. We measured the snakes using a dial caliper (Mitutoyo 505–672 D20tn). Due to their small size, S. dekayi could be easily handled by a single investigator when taking measurements. Eight linear measurements were taken: snout-vent length (SVL), tail length (TL), internostril (IN), interocular (IO), jaw length (JL), head length (HL), head height (HH), and head width (HW; Fig. 2A). We also counted the ventral scales (VS) and subcaudal scales (SS) of each snake. We did not measure morphological parameters when we encountered abnormalities (e.g., no IN measurement for an individual if its snout was injured or deformed). Photos of their dorsal view of the head were taken using a digital camera (Cannon EOS RP or Cannon PowerShot SX70 HS) for geometric morphometric analyses. Snakes were placed on a flat tray for photography. We positioned and partially constrained the snakes by hand to ensure a consistent photography process across different individuals. A portion of the snakes captured in the field were sacrificed as voucher specimens (n = 33) and were measured and photographed immediately after being processed for preservation. Several freshly killed snakes were also found during the sampling period, potentially resulting from human-snake encounters or predation attempts by other urban animals. Since their bodies were all found intact, they were included in the study and measured shortly after being processed for preservation as well. Sex of snakes was first determined by everting hemipenes of live specimens, but to avoid intrusive examination of this small species, if hemipenes were not successfully everted by hand on the first attempt, sex was then determined via VS and SS counts, which are known to exhibit sexual dimorphism in many snakes including S. dekayi (King, 1997). If a snake that we failed to evert hemipenes on was subsequently sacrificed, its sex was then crosschecked by everting hemipenes during the specimen fixation step. All sampling procedures followed a sterile protocol to avoid spreading potential diseases (Rzadkowska et al., 2016) and the Guidelines for Live Amphibians and Reptiles in Field and Laboratory Research by the American Society of Ichthyologist and Herpetologist (available at: https://www.asih.org/resources). The procedures were approved by Rutger University–Newark IACUC (IACUC protocol ID PROTO999900908) and New Jersey Division of Fish and Wildlife (Scientific Collecting and Wildlife Salvage Permits no. SW2017040, SW2018011, SC2019074, SW2019020, SC2020054, SW2020008, SC2021017, and SW2021007).

Figure 2 (A) Six head-dimension measurements taken for the linear morphometric analyses and (B) thirteen landmarks used in the geometric morphometric analyses.

Using natural log-transformed linear morphometric data, we performed analyses of covariance (ANCOVA) with SVL as a covariate to compare head dimensions of S. dekayi from different locations and males and females pooled from different locations. For the interpopulation analyses, Tukey’s post-hoc tests were used to determine the significance of effects between population pairs. Skewness and kurtosis of the head-dimension variables were examined for data normality. Levene’s test was performed to check for variance homogeneity. Because SVL is a sexually dimorphic trait itself in S. dekayi (Ernst & Ernst, 2003), scaling other morphological traits against SVL to detect sexual dimorphism and population differentiation can potentially generate complications that hinders appropriate interpretations (Kratochvíl et al., 2003). To account for this issue, we complemented the above-mentioned analyses by repeating them using HL as a covariate. The statistical analyses were performed using R v4.1.2 (R Core Team, 2021).

We picked 13 landmarks of the dorsal view of the head for geometric morphometric analyses (Fig. 2B). Landmarks were only taken on the left-side of the head to avoid pseudoreplication (Hurlbert, 1984). With the exception of the tip of the snout, all landmarks were based on the junctions of dorsal scales. Landmarks of each specimen were digitized using tpsDig2, v2.31 (Rohlf, 2018). We used generalized Procrustes analysis (GPA) to first align head shape of different individuals. A canonical variates analysis (CVA) was conducted to assess the global population differentiation using R package morpho (Schlager, 2017). The shape changes associated with each canonical variate (CV) were visualized on transformation grids in MorphoJ v1.07a (Klingenberg, 2011). A discriminant analysis with leave-one-out cross-validation was then performed to examine if individuals can be assigned to their own population and sex through a pairwise population assessment in MorphoJ. A Procrustes multivariate analysis of variance (MANOVA) was conducted using the linear model head shape ∼log(head size) + (site * sex) with a randomized residual permutation procedure (RRPP) with 10,000 permutations. Head size was the centroid size of the snake head shape, which was used as a covariate. Site used were NW, NB, PA, FL, and VC. We also conducted post-hoc pairwise least squares mean comparisons with the same number of RRPP random permutations. Additionally, a thin-plate spline analysis was conducted to help visualize how the consensus head shapes of different populations and sexes vary from the global consensus shape on transformation grids. The statistical analyses above, unless mentioned otherwise, were performed using geomorph v4.0.1 (Baken et al., 2021) within R v4.1.2 (R Core Team, 2021).

Results

The six study sites displayed different levels of urbanization (Fig. 3). Among them, MR was the least urbanized site with only 0.30% imperviousness. The sites VC and NB were moderately urbanized with 35.53% and 36.02% imperviousness respectively; FL and PA were highly urbanized with 52.10% and 55.56% imperviousness respectively. The NW site was the most urbanized site with a 79.16% imperviousness. With the percent imperviousness ranging from a low of 0.30% to a high of 79.16%, the study sites were representative of a complete rural–urban gradient within the study region.

Figure 3 Four study sites with increasing levels of urbanization from left to right.

(A) Maurice River (% imperviousness = 0.30), (B) New Brunswick (% imperviousness = 36.02), (C) Fort Lee (% imperviousness = 52.10), and (D) Newark (% imperviousness = 79.16).

The linear morphometric data of 157 individuals of S. dekayi were obtained from six populations: NW (n = 33), NB (n = 26), PA (n = 31), FL (n = 32), MR (n = 9), and VC (n = 26). This included 73 females, 74 males, and 10 individuals with undetermined sex that were only used in the interpopulation analyses. When using SVL as a covariate, the ANCOVA indicated that location significantly affected interpopulation morphometric differentiation in terms of HH and HW (HH: F5,136 = 3.073, P = 0.012; HW: F5,149 = 2.677, P = 0.024; Fig. 4). However, Tukey’s post-hoc tests showed that only the HH difference between snakes from VC and those from four other populations, FL, NB, NW, and PA were significant (VC-FL: P = 0.040; VC-NB: P = 0.016; VC-NW: P = 0.003; VC-PA: P = 0.050), and only the HW difference between snakes from VC and NB were significant (P = 0.006). When using HL as a covariate, the ANCOVA indicated that location significantly affected interpopulation morphometric differentiation regarding all the examined traits besides JL (IN: F5,148 = 2.680, P = 0.024; IO: F5,148 = 3.530, P = 0.005; HH: F5,136 = 4.092, P = 0.002; HW: F5,148 = 2.833, P = 0.018; Fig. S1). Tukey’s post-hoc tests showed similar patterns for HH (VC-FL: P = 0.007; VC-NB: P = 0.012; VC-NW: P < 0.001; VC-PA: P = 0.022) and HW (VC-NB: P = 0.012; VC-NW: P = 0.016), while the significant IN differentiation was attributed to the population pair of NB-NW and that of IO was attributed to the population pairs of NB-NW and NB-PA.

Figure 4 Head morphology difference displayed among six Storeria dekayi populations in terms of their (A) head height (F = 3.073, P = 0.012) and (B) head width (F = 2.677, P = 0.024).

When using SVL as a covariate, S. dekayi displayed sexual dimorphism in terms of IN, IO, JL, and HL. Males exhibited a wider IN (F1,142 = 17.889, P < 0.001), a wider IO (F1,142 = 22.185, P < 0.001), a larger JL (F1,143 = 12.123, P < 0.001), and a larger HL (F1,144 = 29.208, P < 0.001) than females (Fig. 5). When using HL as a covariate, S. dekayi displayed sexual dimorphism in terms of JL, HH, and HW. Females exhibited a larger JL (F1,143 = 6.482, P = 0.012), a larger HH (F1,131 = 7.797, P = 0.006), and a larger HW (F1,142 = 19.415, P < 0.001) than males (Fig. S2).

Figure 5 Storeria dekayi displays head shape sexual dimorphism.

Males have a longer (A) internostril distance (F = 19.437, P < 0.001), (B) a longer interocular distance (F = 25.578, P < 0.001), (C) a longer jaw (F = 12.952, P < 0.001), and (D) a longer head (F = 29.341, P < 0.001) than the females.

The geometric morphometric data of 144 individuals of S. dekayi were obtained from five populations: NW (n = 20), NB (n = 34), PA (n = 25), FL (n = 44), and VC (n = 21). This included 56 females, 66 males, and 22 individuals with undetermined sex that were not used in analyses concerning sexual dimorphism. The CVA discovered four distinct canonical variates. The first three CVs accounted for 91.23% of the variance. When being visualized on the scatterplots, there was little overlapped space between the most urbanized NW and the other populations along CV1, but overall, the first three CVs did not clearly distinguish the five populations (Fig. 6). The main head shape changes associated with CV1 were explained by the shortening of the snout tip and the expansion of both anterior and posterior ends of the frontal scale; the main head shape changes associated with CV2 were explained by narrowing of the head and shrinking of the posterior end of the parietal scale; the main head shape changes associated with CV3 were explained by the shape change of the supraocular scale (Fig. 7). The discriminant analysis indicated that most individuals could be assigned to their correct populations (>85% accuracy), but after the cross-validation, acceptable assignments (>80% accuracy) only occurred when comparing the population pairs of FL-PA and NW-PA, in which all populations involved were from the three most urbanized sites (Table 1). According to the discriminant analysis, 71.2% males and 71.4% females could be correctly assigned after the cross-validation, which were relatively low rates.

Figure 6 A scatterplot of the first two canonical axes for the head dimension of five Storeria dekayi populations.

Ellipses displayed are 80% confidence ellipses.

Figure 7 Transformation grids showing the Storeria dekayi head dimension changes associated with the first three canonical variates.

Table 1 Results of pairwise population assignment after a leave-one-out cross-validation.

Allocated to	
True group		New Brunswick	Fort Lee	% Correct	
New Brunswick	24	10	70.6	
Fort Lee	12	32	72.7	
	New Brunswick	Newark	% Correct	
New Brunswick	29	5	85.3*	
Newark	8	12	60	
	New Brunswick	Perth Amboy	% Correct	
New Brunswick	25	9	73.5	
Perth Amboy	10	15	60	
	New Brunswick	Van Cortlandt	% Correct	
New Brunswick	25	9	73.5	
Van Cortlandt	7	14	66.7	
	Fort Lee	Newark	% Correct	
Fort Lee	38	6	86.4*	
Newark	5	15	75	
	Fort Lee	Perth Amboy	% Correct	
Fort Lee	37	7	84.1*	
Perth Amboy	5	15	80*	
	Fort Lee	Van Cortlandt	% Correct	
Fort Lee	37	7	84.1*	
Van Cortlandt	10	11	52.4	
	Newark	Perth Amboy	% Correct	
Newark	16	4	80*	
Perth Amboy	5	20	80*	
	Newark	Van Cortlandt	% Correct	
Newark	17	3	85*	
Van Cortlandt	5	16	76.2	
	Perth Amboy	Van Cortlandt	% Correct	
Perth Amboy	22/17	3/8	68	
Van Cortlandt	1/7	20/14	66.7	
Notes.

* acceptable assignment proportion (>80%).

Table 2 Procrustes MANOVA statistics based on a randomized residual permutation procedure (RRPP) with 10,000 random permutations.

	df	R 2	Z	P	
log(head size)	1	0.1877	6.6070	0.0001*	
Site	4	0.0950	6.2109	0.0001*	
Sex	1	0.0228	3.1870	0.0009*	
Site × Sex	4	0.0336	1.4547	0.0715	
Notes.

* Value is significant at α = 0.05.

Procrustes MANOVA indicated that there were significant interpopulation differences and sexual dimorphism for the head shape of S. dekayi (Table 2). The post-hoc pairwise comparisons suggested that the head shape of S. dekayi from different locations were all significantly different from each other when shape allometry and sexual dimorphism were accounted for; the pairwise least squares mean distance generally increased with the urbanization level difference between the pair of sites (Table 3). After accounting for shape allometry, sexual dimorphism regarding head shape was insignificant in NW, PA, and VC, two of which were from the most urbanized sites. It was also hard to distinguish PA females from NB and FL males, and VC females from NB and PA males (Table 4). The thin-plate spline analysis did not reveal any distinct visualizable head shape differences between the five populations and the global consensus (Fig. 8), or that between the two sexes and the global consensus (Fig. 9).

Table 3 Pairwise least squares mean distances and P-values based on 10,000 random permutations using RRPP associated with the Procrustes MANOVA in Table 2.

The null model for RRPP was log(size) + sex. The full model was log(size) + sex× site.

	NB	FL	NW	PA	VC	
NB	–	0.0005*	0.0001*	0.0290*	0.0171*	
FL	0.0231	–	0.0001*	0.0018*	0.0005*	
NW	0.0293	0.0272	–	0.0130*	0.0003*	
PA	0.0177	0.0203	0.0207	–	0.0004*	
VC	0.0214	0.0232	0.0309	0.0244	–	
Notes.

Abbreviations NB New Brunswick

FL Fort Lee

NW Newark

PA Perth Amboy

VC Van Cortlandt Park

Values below diagonal are distances; values above diagonal are P-values.

* Value is significant at α = 0.05.

Table 4 Pairwise least squares mean distances and P-values based on 10,000 random permutations using a randomized residual permutation procedure (RRPP) associated with the Procrustes MANOVA in Table 2.

The null model for RRPP was log(size). The full model was log(size) + sex× site.

	NB female	FL female	NW female	PA female	VC female	NB male	FL male	NW male	PA male	VC male	
NB female	–	0.0023*	0.0002*	0.0040*	0.2100	0.0271*	0.0008*	0.0001*	0.0263*	0.0033*	
FL female	0.0272	–	0.0141*	0.0157*	0.0220*	0.0015*	0.0492*	0.0001*	0.0048*	0.0033*	
NW female	0.0401	0.0303	–	0.3183	0.0194*	0.0048*	0.0167*	0.0518	0.0287*	0.0010*	
PA female	0.0292	0.0255	0.0223	–	0.0217*	0.0656	0.0579	0.0324*	0.2628	0.0026*	
VC female	0.0234	0.0281	0.0358	0.0317	–	0.1220	0.0072*	0.0083*	0.0591	0.2295	
NB male	0.0247	0.0294	0.0345	0.0244	0.0258	–	0.0459*	0.0266*	0.6721	0.0358*	
FL male	0.0309	0.0207	0.0315	0.0239	0.0315	0.0236	–	0.0111*	0.0106*	0.0115*	
NW male	0.0362	0.0365	0.0273	0.0259	0.0339	0.0258	0.0275	–	0.0211*	0.0017*	
PA male	0.0240	0.0252	0.0300	0.0195	0.0260	0.0148	0.0241	0.0259	–	0.0102*	
VC male	0.0297	0.0273	0.0389	0.0322	0.0223	0.0255	0.0252	0.0335	0.0253	–	
Notes.

Abbreviations NB New Brunswick

FL Fort Lee

NW Newark

PA Perth Amboy

VC Van Cortlandt Park

Values below diagonal are distances; values above diagonal are P-values.

* Value is significant at α = 0.05.

Figure 8 Transformation grids of the five Storeria dekayi populations warped from the consensus alignment of all individuals examined.

Figure 9 Transformation grids of male and female Storeria dekayi warped from the consensus alignment of all individuals examined.

Discussion

With the accelerating pace of global urbanization, it is important to understand the various ecological effects, whether harmful in the long term or not, urban environments can potentially cause on the urban-dwelling wildlife, such as the changes of their morphology. In this study, we investigated morphological differentiation and sexual dimorphism of head shape in a small North American snake, Storeria dekayi, across a rural–urban gradient. Our results provide important insights regarding how urban-dwelling fauna respond to this novel ecosystem.

Through examining six S. dekayi populations inhabiting sites varying in levels of urbanization, we found evidence of interpopulation morphological differences. Linear morphometrics suggested that S. dekayi morphology across sites were differentiated in four traits: internostril distance, interocular distance, head height, and head width. However, as Tukey’s post-hoc tests indicated, the head height and head width differences were mainly attributed to the clear distinction between the Van Cortlandt Park (VC) individuals and the other populations. Results concerning VC individuals should be interpreted with caution, as VC specimens used in this study were all old voucher specimens preserved in ethanol and collected from 1939 to 1944, while all the other individuals were live, freshly killed, or newly sacrificed specimens sampled during the study period. Accordingly, it is possible that the distinct difference between VC and the other individuals was merely an artifact resulting from the older state of the historical voucher specimens. In addition, the percent imperviousness calculated in this study only reflected the current level of urbanization at the study sites, while all VC samples were collected more than 80 years ago when relevant urbanization data were unavailable. Although Van Cortlandt Park was established more than 50 years prior to the collecting period and large pieces of woodlands persist in the park throughout the years, it had undergone a series of drastic changes in the early 20th century (Henning, 2015). Hence, the urbanization level in this area was likely very different and was experiencing constant fluctuations at the time, which had potential to induce rapid morphological shifts of the snake population in the park.

More concrete proof of interpopulation morphological differentiation and its correlation with urbanization levels, on the other hand, was revealed by the Procrustes MANOVA. The post-hoc pairwise least squares mean distance values in general agreed with the rural–urban gradient reflected among the five sites in the analyses. Further, as inferred from other geometric morphometric analyses, populations dwelling in highly urbanized sites were likely more divergent morphologically than the rest. According to the CVA results, only the population from the most urbanized site, Newark (NW), was morphologically separated from those inhabiting the other sites. Based on the transformation grid visualizations, this could potentially be explained by a shorter but bulkier head the population exhibited. Such head shape could be associated with the diet of this urban population. Because the ability and efficiency of snakes to swallow larger prey items are limited by their head and gape size, species preying on larger preys tend to have wider and thicker heads (Wüster, Duarte & Salomão, 2005; Hampton, 2011; Fabre et al., 2016). This also applies intraspecifically, as demonstrated by the fact that snake populations inhabiting regions with more large-bodied prey having larger heads (Forsman, 1991; Fabien et al., 2004; Brecko et al., 2011). Likely, the same could be used to explain the divergence of the NW population. Nonnative prey of S. dekayi, such as exotic earthworms, often colonize and displace their native counterparts in human-altered habitats (Hendrix & Bohlen, 2002; Ziter et al., 2021). As such, the trait of having a bulkier head might be favored in highly urbanized sites due to the dominance of large-bodied introduced preys like the Eurasian lumbricid worms.

Discriminant analysis after cross-validation only showed clear distinction of head shape among populations from the three most urbanized sites, Fort Lee (FL), Perth Amboy (PA), and NW, which could be an indication of enhanced and more divergent morphological variations among the urban populations. Enhanced morphological variations were found in urban Anolis lizards owing to their niche expansion in the highly heterogenous urban environments (Falvey et al., 2020). However, due to its limited locomotion ability and semi-fossorial natural history (Ernst & Ernst, 2003), it is unlikely that S. dekayi would take advantage of this environmental heterogeneity. In fact, urban populations like those in NW were more likely to experience a more homogenous habitat than their rural counterparts, as they were confined in small green spaces with depauperate vegetation. Thus, site-specific directional selection is more likely to explain the divergent pattern observed. For instance, the PA site was characterized by an abandoned railway running across the habitat, and the NW site was dominated by planted vegetation, which were both unique anthropogenic factors not shared by the other sites. However, as for whether such site-specific features could indeed affect morphological changes of S. dekayi, further investigation would still be required.

Storeria dekayi is sexually dimorphic, with evidence clearly shown in the linear morphometric results. Sex differences in internostril distance, interocular distance, jaw, and head length all indicate a proportionally larger head of males than females, which aligns with the results from early studies (Shine, 1991a; King, 1997). Sexual dimorphism of snakes is mostly shown in the form of one sex exceeding the other in both body and head sizes, which is considered to be associated with dietary divergence regarding the prey size (Camilleri & Shine, 1990; Shine, 1991a). However, as males of S. dekayi have a larger relative head size and a smaller relative body size than females, the resulting absolute head size can be similar to that of females, leading to a potential dietary convergence with unclear adaptive benefits (King, 1997).

Alternatively, the larger relative head size of males might be explained by their different sexual role. Previous studies have reported sexually dimorphic head shape on various elapid, viperid, and natricid snakes (Andjelković, Tomović & Ivanović, 2016; Borczyk et al., 2021; Borczyk, Puszkiewicz & Bury, 2024). However, such dimorphism primarily occurs in the frontal and parietal regions (Borczyk, 2023). The frontal and parietal regions of the snake head house various sensory organs, such as the olfactory bulb, the vomeronasal organs, and the eyeballs, which are all involved in the mate searching behaviors of males (Andrén, 1982; Shine et al., 2005; Mason & Parker, 2010). Male S. dekayi are known to follow pheromone trails of females actively during their breeding season (Ernst & Ernst, 2003). Thus, their vomeronasal organs, the organs responsible for pheromone detection, are likely to be more developed than females, resulting in a longer internostril distance (Borczyk, Puszkiewicz & Bury, 2024). Although we did not measure the eye size directly, to facilitate mate searching, males S. dekayi might also possess more developed eyeballs, which can lead to a longer interocular distance (Borczyk et al., 2021; Borczyk, Puszkiewicz & Bury, 2024). However, it is still noticeable that when head-dimension variables were scaled against head length to avoid the exaggeration of already sexually dimorphic snout-vent length (Kratochvíl et al., 2003), S. dekayi exhibited sexual dimorphism in a different set of traits, and females showed greater values than males for all those traits. Such completely different results indicated that the perceived sexual dimorphism found when using snout-vent length as a covariate might not be that pronounced and requires further examination.

When the factor of urbanization level was taken into consideration, contrary to expectation, individuals from the most urbanized sites, such as NW and PA, were generally less sexually dimorphic than the others. This result is consistent with the findings by Bury & Zając (2020), who reported reduced sexual size dimorphism in urban Natrix natrix populations. The authors attributed this to the low resource availability in urban areas, which affected the large-bodied females more severely than small-bodied males and thus closed the size gap between them (Bury & Zając, 2020). Food sources of S. dekayi in urban areas, such as earthworms, may be scarce and depauperate (Xie et al., 2018; Tóth et al., 2020). Consequently, a sexually asymmetrical impact on body size might also explain the reduced sexual dimorphism observed in S. dekayi. Additionally, given the high densities of S. dekayi in urban habitats (Noble & Clausen, 1936; Ernst & Ernst, 2003; Huang, Morin & Ruane, 2025), it is also possible that the lack of sexual dimorphism was an indicator of intensified competition. Specifically, as males have a proportionally larger head, lower intensities of sexual size dimorphism could potentially result in the partition of the absolute head size between the two sexes, alleviating the competition between the two sexes through dietary divergence. Similar sex-by-site interaction was previously documented in S. dekayi when comparing different island and mainland populations, though the factor of urbanization was not considered and no clear trend was found (King, 1997).

Our study sheds light on the morphological trends of the snake Storeria dekayi across the rural–urban gradient, a largely unrepresented taxon in the field of urban ecology due to its secretive nature. By measuring mostly live and freshly sacrificed specimens, this work also has the advantage of reflecting the true head dimensions and underlying muscular structures of the organism compared to similar work that predominately relies on preserved voucher specimens (e.g., Shine, 1991a; Brecko et al., 2011; Ruane, 2015). However, we recognize the limitations of this work and consider the results only preliminary, necessitating further evaluation. Although individuals from the more urbanized sites seemed to be morphologically more distinct and diverse, strong evidence for this and its correlation with urbanization levels is still lacking. While as evidenced by common garden and genetic study results, urban morphological changes can often be attributed to the rapid local adaptive evolution (Badyaev et al., 2008; Winchell et al., 2016; Winchell et al., 2023), without the underlying mechanisms of the observed patterns being directly examined here, we cannot exclude the possibility of non-evolutionary adaptive responses, like acclimation, or even completely non-adaptive and stochastic processes such as genetic drift, leading to the observed results. Furthermore, with snakes of different age classes included in our samples, our work only took the factor of size, but not ontogeny into account due to the relatively small sample size. This could confound the trends observed both within and between populations, because head shape of snakes can change ontogenetically (Vincent, Herrel & Irschick, 2004; López, Manzano & Prieto, 2013; Ruane, 2015). That said, we believe this work is a critical step in understanding urban morphological shifts of secretive organisms like snakes, and encourage future efforts to integrate more advanced techniques along with different types of data and larger sample sizes to provide a more comprehensive view of this phenomenon.

Supplemental Information

Supplemental Information 1 Head morphology differences among six Storeria dekayi populations when using head length as a covariate in the ANCOVA

(A) Internostril (F = 2.680, P = 0.024), (B) interocular (F = 3.530, P = 0.005), (C) head height (F = 4.092, P = 0.002), and (D) head width (F = 2.833, P = 0.018).

Supplemental Information 2 Storeria dekayi displays head shape sexual dimorphism when using head length as a covariate in the ANCOVA

Females have (A) a longer jaw (F = 6.482, P = 0.012), (B) a larger head height (F = 7.797, P = 0.006), and (C) a wider head (F = 19.415, P < 0.001) than males.

We thank E. Rittmeyer, J. Bernstein, and undergraduate volunteers from Rutgers University–New Brunswick and New Jersey Institute of Technology (M. Fukura, R. Gurland, A. Mazza, C. Randik, and Y. Zou) for assisting with lab and field work. We also thank H. John-Alder (Rutgers University–New Brunswick) for helpful comments on the manuscript.

Additional Information and Declarations

Competing Interests

Author Contributions

Animal Ethics

Field Study Permissions

Data Availability

The authors declare there are no competing interests.

Tianqi Huang conceived and designed the experiments, performed the experiments, analyzed the data, prepared figures and/or tables, authored or reviewed drafts of the article, and approved the final draft.

Peter J. Morin conceived and designed the experiments, authored or reviewed drafts of the article, and approved the final draft.

Sara Ruane conceived and designed the experiments, authored or reviewed drafts of the article, and approved the final draft.

The following information was supplied relating to ethical approvals (i.e., approving body and any reference numbers):

This work was approved by Rutger University Newark Institutional Animal Care and Use Committee ID PROTO999900908.

The following information was supplied relating to field study approvals (i.e., approving body and any reference numbers):

This work was conducted under New Jersey Division of Fish and Wildlife Scientific Collecting and Wildlife Salvage Permits no. SW2017040, SW2018011, SC2019074, SW2019020, SC2020054, SW2020008, SC2021017, and SW2021007.

The following information was supplied regarding data availability:

The data is available at Zenodo: Anonymous, A. (2025). Data for Storeria dekayi morphology study [Data set]. Zenodo. https://doi.org/10.5281/zenodo.14984017.

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
