# Peer review of "Interpopulation morphological differences and sexual dimorphism of Dekay’s brownsnake (Storeria dekayi) along a rural–urban gradient"

_PeerJ, doi:10.7717/peerj.19439_

## Round 0.1 · original submission · Major Revisions

Thank you very much for your manuscript titled “Interpopulation Morphological Differences and Sexual Dimorphism of Dekay’s Brownsnake (Storeria dekayi) Along a Rural-Urban Gradient.” that you sent to PeerJ.

This study presents very valuable and relevant information on interpopulation and sexual differences using both linear and geometric morphometric analyses. The manuscript is well-written and follows methodological rigor. As you will see below, comments from referee 1 suggest a major revision, while reviewers 2 and 3 suggest a minor revision before your paper can be published. Given this, I would like to see a major revision dealing with the comments. I will be happy to accept your article pending further revisions, detailed by the referees, which largely focus on clarify several methodological and interpretation doubts and discuss the results.

Reviewer 1 suggests clarify methodological doubts and those related to the discussion of the results. S/he also has doubts about the validity of the sample sizes.

Reviewer 2's observations are noted directly at each point in the manuscript, trying to improve its clarity.

Reviewer 3 suggest some bibliographical references that may contribute positively to the manuscript, especially in the discussion.

Please note that we consider these revisions to be important and your revised manuscript will likely need to be revised again.

Reviewer 1 ·

Basic reporting

I am not a native English speaker.
Literature references, sufficient field context provided. YES.
Professional article structure. YES
Self-contained with relevant results to hypotheses. PARTIALLY. THE AUTHORS ADMIT SOME WEAKNES OF THE WORK AT THE END OF THEIR PAPER

Experimental design

YES.
PARTLY. HE AUTHORS ADMIT SOME WEAKNES OF THE WORK AT THE END OF THEIR PAPER
YES BUT NOT IN QUANTITATIVE TERMS
YES

Validity of the findings

NOTHING TO ADD
NOTHING TO ADD
NOTHING TO ADD

Additional comments

Review of
Interpopulation morphological differences and sexual dimorphism of Dekay’s brownsnake (Storeria dekayi) along a rural-urban gradient (#110852)
SOME SUGGESTIONS FOR IMPROVING OF THE MANUSCRIPT:
I am not a native English speaker so I cannot make any statements about the language.
Introduction and background show appropriately the context. Literature well referenced and relevant.
33 Introduction 34 Habitat and environmental change can influence the morphology of organisms REPHRASE: HABITAT IS PART OF ENVIRONMENT
With rapid 36 urbanization occurring in recent decades, drastic morphological shift would be expected on organisms living in the novel urban environments. CLEARIFY THE TERM RAPID IN AN EVOLUTIONARY ADAPTATIVE CONTEXT.
ALL THE NOT ORIGINAL TEXTS SHOULD HAVE CITATIONS AND FINAL PAPER REFERENCES.
88 Dekays brownsnake (Storeria dekayi) is a small, semi-fossorial snake species that 89 inhabits a wide variety of terrestrial habitats, but is most commonly found in anthropogenically 90 disturbed and urbanized environments EXEMPLIFY PLEASE.
HIGHLIGHT AND EXPLAIN MORE THE LIMITATIONS OF YOUR APPROACH AND YOUR STUDY RESULTS PLEASE. HOW THIS WORK SHOULD TO BE CONTINUE.
THE LOW NUMBER OF SAMPLING SITES INFLUENCE THE AQURACY OF THE RESULTS AND CONCLUSIONS?
THE RELATIVELY SMALL AREAS OF SAMPLING RELATED TO THIS SPECIES DISTRIBUTION RANGE NFLUENCE THE AQURACY OF THE RESULTS AND CONCLUSIONS?
DESCRIBE SHORTLY THE HABITATS CHARACTERISTICS OF THE SAMPLING SITES.
WHY THE USED MEASURES WERE SELECTED AND NOT MORE OR OTHERS?
EXPLAIN ABREVIATIONS WHERE FIRST APPEAR IN THE TEXT.
PLEASE HIGHLIGHT AND CONCLUDE WHY THE AUTHORS CONSIDER THE REGISTERED MORPHOLOGIC VARIATIONS AS BEING NOT A RATHER NATURAL MORPHOLOGICAL VARIATION INCLUDING BETWEEN SEXES AND NOT AN INFLUENCE OF RURAL AND URBAN LIKE ENVIRONMENTS

·

Basic reporting

no comment

Experimental design

no comment

Validity of the findings

no comment

Additional comments

The manuscript titled “Interpopulation Morphological Differences and Sexual Dimorphism of Dekay’s Brownsnake (Storeria dekayi) Along a Rural-Urban Gradient” explores how urbanization affects the morphology and sexual dimorphism of Storeria dekayi, a snake species inhabiting rural and urban areas in the United States. The study employs both linear and geometric morphometric analyses to assess interpopulation and sexual differences. The manuscript is well-written and follows methodological rigor. I recommend its acceptance with minor revisions to improve clarity and impact.
Suggested Revisions:
• Introduction (Line 88): The authors should include the family or subfamily of Storeria dekayi. Additionally, they should mention the species' geographic distribution and whether it exhibits intraspecific variation (e.g., subspecies) across its range.
• Materials and Methods (Starting at Line 105): The sampling methodology needs further clarification. Specifically, the authors should describe how specimens were collected. Were traps used, or was an active search method (e.g., time-constrained searches) employed? Were encounters incidental? Additionally, how much time was spent searching in each of the five study areas?
• Line 108: The authors should provide the size of each study area.
• Figure 1: I recommend including a small inset map showing the broader geographic context of the study site, such as the United States, with the locations of New York and New Jersey highlighted.
• Line 118: It should be clarified whether the photographs for geometric morphometric analysis were taken in the field immediately after capture. If so, it is important to specify how the snakes were immobilized during photography.
• Line 118 (continued): The authors state that some snakes were released. Were these individuals marked using a specific method? Was there any attempt at recapture?
• Line 126: The authors should specify how many specimens were captured and sacrificed, how many were found dead, and how many were released.
• Discussion (Lines 223–228): The first paragraph should either be removed or incorporated into the Introduction.
• Lines 306–307: The sentence “However, we … further evaluation” should be removed, as the manuscript presents robust results that do not require additional justification.
• Lines 319–322: The final sentence “That said, … of this phenomenon” should also be removed for conciseness.
Overall, this study makes a valuable contribution to understanding how urbanization influences snake morphology and sexual dimorphism. These minor revisions will enhance the clarity and presentation of the manuscript.

Reviewer 3 ·

Basic reporting

I find your paper very interesting and important. However, I think some issues could be improved. Here I list my main assertation:

The SVL is dimorphic in most of snake species. Thus, scaling other traits against SVL may affect the results and lead to false conclusions on presence (or lack) of sexual dimorphism. Please see Kratochwil et al. 2003: Misinterpretation of character scaling: a tale of sexual dimorphism in body shape of common lizards. Candian Journal of Zoology 81: 1112-1117. I strongly suggest to you to do ANCOVA on head dimension with Head Length as covariate in addition to already presented results.

Discussion should be expanded. Please addres there to the results of Bury and Zając, who studied Natrix natrix. You mention them in the Introduction section, but not in the discussion.

Please add some discussion on the observed sexual dimorphism. In my opinion, dimorphism in IN and IO is specially interesting, since it is rarely mentioned. It has been found in the skull structures corresponding to the head dimensions you measured in Aipysurus and Laticauda (Borczyk et al. 2021: Sexual dimorphism and skull size and shape in the highly specialized snake species, Aipysurus eydouxii (Elapidae: Hydrophiinae), PeerJ 9:11311; Borczyk 2023: Sexual dimorphism in skull size and shape of Laticauda colubrina (Serpentes: Elapidae). PeerJ 11: e16266) and in heads of two Vipera species (Borczyk 2024: Sexual dimorphism and allometry in the head and body size of two viperid snakes (genus Vipera), Belgian Journal of Zoology 154: 31-44)

Experimental design

No comment

Validity of the findings

Please add discussion on sexual dimorphims in the head shape you observe in your species. It is importnat but not covered enough in you discussion.

Additional comments

Please correct "ZajĄc" - there should be "Zając"
Figures are of good quality. However, I suggest to provide an additional one showing the scheme od linear head measurements along with more precise description of the dimensions measured. For example, Head length is usually taken from the tip of the snout to the rear end of mandible, but in some cases there might be to the end of pileus etc.

---

## Round 0.2 · accepted · Accept

After reviewing this revised version of your manuscript, I see that the main comments suggested by the reviewers have been included, while the suggestions not considered are justified in detail. Therefore, I am satisfied with the current version and consider it ready for publication.

·

Basic reporting

no comment

Experimental design

no comment

Validity of the findings

no comment

Additional comments

I have completed my re-review of the manuscript. After carefully evaluating the revisions made by the authors and their responses in the rebuttal letter, I find the manuscript suitable for publication in PeerJ.

Reviewer 3 ·

Basic reporting

Dear authors,
I have read your revised article and the responses to the earlier comments carefully. I think the article is well written and contains important and valuable observations. I have no further comments on the text. I look forward to the final, published version of your article. Congratulations on your interesting research!

Experimental design

Nothing to add

Validity of the findings

Nothing to add

Additional comments

Nothing to add